# Gel phase formation in dilute triblock copolyelectrolyte complexes

Samanvaya Srivastava[1,2], Marat Andreev[1], Adam E. Levi[1], David J. Goldfeld[1], Jun Mao[1,2], William T. Heller[3], Vivek M. Prabhu[4], Juan J. de Pablo[1,2] & Matthew V. Tirrell[1,2]

Assembly of oppositely charged triblock copolyelectrolytes into phase-separated gels at low polymer concentrations ($<$1% by mass) has been observed in scattering experiments and molecular dynamics simulations. Here we show that in contrast to uncharged, amphiphilic block copolymers that form discrete micelles at low concentrations and enter a phase of strongly interacting micelles in a gradual manner with increasing concentration, the formation of a dilute phase of individual micelles is prevented in polyelectrolyte complexation-driven assembly of triblock copolyelectrolytes. Gel phases form and phase separate almost instantaneously on solvation of the copolymers. Furthermore, molecular models of self-assembly demonstrate the presence of oligo-chain aggregates in early stages of copolyelectrolyte assembly, at experimentally unobservable polymer concentrations. Our discoveries contribute to the fundamental understanding of the structure and pathways of complexation-driven assemblies, and raise intriguing prospects for gel formation at extraordinarily low concentrations, with applications in tissue engineering, agriculture, water purification and theranostics.

[1] Institute for Molecular Engineering, The University of Chicago, Chicago, Illinois 60637, USA. [2] Institute for Molecular Engineering, Argonne National Laboratory, Lemont, Illinois 60439, USA. [3] Biology & Soft Matter Division, Oak Ridge National laboratory, Oak Ridge, Tennessee 37831, USA. [4] Material Measurement Laboratory, National Institute of Standards and Technology, Gaithersburg, Maryland 20899, USA. Correspondence and requests for materials should be addressed to J.J.de.P (email: depablo@uchicago.edu) or to M.V.T. (email: mtirrell@uchicago.edu).

Polyelectrolyte complexation—associative phase separation of oppositely charged polyelectrolytes in aqueous milieu—can be harnessed via intelligent macromolecular design to drive self-assembly of nanoscale micelles with highly hydrated cores[1–4]. Electrostatic interactions between oppositely charged chains lead to entropy gains from the release of the counterions associated with the chains that drive complexation[5,6]. Conjugation of either or both polyelectrolytes with a neutral polymer block allows for molecular tuning of the microphase separation and results in micelles of diverse shapes[7–9], including spherical[10–12], Janus[13–15], rod-like[16–18] and vesicular micelles[19,20]. Substantial water content[21–23] and extremely low surface tension against water[24–26] of liquid polyelectrolyte complexes facilitate encapsulation of hydrophilic small molecule drugs[27], nucleic materials[11,28,29] and proteins[30–32] in the micelle cores, enabling polyelectrolyte complex (PEC) micelle carriers for targeted, encapsulated delivery of hydrophilic cargos[29,33–40].

Increasing micelle concentrations above the overlap concentration leads to jamming of micelles resulting in highly hydrated, viscoelastic solid materials[41–48]. An evolution of ordered micellar arrangements and morphology with increasingly denser packings, akin to those resulting from jamming of uncharged, amphiphilic block copolymer micelles[49–51] have been reported[42,45]. Specifically, a disorder-order transition, exemplified by ordering of spherical PEC domains into a cubic lattice, followed by morphological transitions of PEC domains from spheres to hexagonally close-packed cylinders to parallel-stacked lamella[42,45,48] are observed with increasing micelle concentrations, allowing for facile tuning of the gel properties[44,45,47,52,53]. In addition, growth factors, nutrients and drugs can easily be encapsulated in the well-hydrated complex cores[32,53], and the electrostatic cross-links render excellent self-healing[45] properties, as well as a swift responsiveness to variations in salt concentration and pH (refs 42,43,45), making these PEC hydrogels extremely attractive materials as tissue growth supports[53] and bioadhesives.

Complexation-driven self-assemblies are understood to proceed in an analogous manner to uncharged, amphiphilic block copolymers, despite the stark differences among the driving forces for self-assembly in each case. In this article, we report the discovery of spontaneous interconnected gel-phase formation on solvation of oppositely charged ABA triblock copolyelectrolytes, and phase separation of the gel phases from the solution at very low polymer concentrations. An observable phase of individual flower-like micelles is absent. The gel formation is driven by the enhanced propensities of the neutral midblocks to form bridges between the PEC domains. The reported observations illustrate the long-range structure-directing contributions of electrostatic forces, in addition to their role in actuating complexation. AB diblock copolyelectrolytes, expectedly, form discrete star-like micelles at low polymer concentrations. Furthermore, we demonstrate that at high-polymer concentration, both AB diblock and ABA triblock copolyelectrolytes assemblies have near identical scattering signatures; however, the evolution of scattering patterns with increasing polymer concentrations follow distinct pathways. Our findings mark a significant departure from the established uncharged, amphiphilic block copolymer-like assembly mechanisms for complexation driven assemblages, and will have implications in guiding the design principles for PEC gels for many applications such as cell scaffolds for tissue engineering, charge based flocculating agents for water purification, thin nutrient films for agriculture and extremely sensitive theranostic probes.

## Results

**Structure and the disorder-order transition in PEC gels.** Figure 1 highlights the similarities and differences among the structures formed via self-assembly of pairs of oppositely charged di- and triblock copolyelectrolytes comprising poly(ethylene oxide) (PEO) and either guanidinium chloride (cationic) or sodium sulfonate (anionic) functionalized poly(allyl glycidyl ether) (PAGE) blocks. The scattering intensity ($I(q)$, $q$ is the wave vector) profiles from both di- and triblock copolyelectrolyte assemblies, shown in Fig. 1a, indicate incipient ordering in the structures at high-polymer concentration $\phi$ (10% by mass), with strong correlations among the PEC domains leading to prominent primary peaks in the $I(q)$ profiles. Each of the oppositely charged pairs of polyelectrolytes studied were functionalized from the same PEO–PAGE block copolymer and were mixed in charge equivalent amounts. Therefore, any effects of charge or length mismatch on the self-assembly were eliminated, thus producing strongly phase segregated model assemblies. The diblock polymers were precisely synthesized to be nearly half the size of the triblock copolymers; therefore, the morphologies and arrangements of the PEC domains at high concentrations, where the PEO coronas began to impinge, regardless of whether these tethered corona chains were brushes, loops or bridges, were expected to be near identical, leading to almost indistinguishable $I(q)$ profiles. The faintly sharper peaks in diblock copolyelectrolyte gels could be attributed to fewer topological constraints to self-assembly[47]. Furthermore, salt is known to affect the nature of such electrostatics-driven assemblies. However, the results reported here describe studies carried out with no added salt; detailed investigations of salt effects on the structure of similar materials systems have been reported earlier[45–47].

Ordering of the PEC domains on a body-centred cubic lattice was observed on increasing $\phi$ from 10 to 40% by mass gels in both the material systems, driven by compression of the neutral midblock[42,45], which form swollen interstices between the complex domains. The scattering patterns for the 40% by mass samples consist of scattering from strongly correlated PEC cores in conjunction with the sharp primary ($q^* \sim 0.033\,\text{Å}^{-1}$), secondary ($\sqrt{2}q^* \sim 0.046\,\text{Å}^{-1}$) and tertiary ($\sqrt{3}q^* \sim 0.055\,\text{Å}^{-1}$) Bragg reflection peaks, corresponding to an arrangement of the PEC cores in the body centred cubic (BCC) lattice. Such ordering transitions have been reported earlier for similar material systems[45,46,48]. Last, $I(q)$ grew at low $q$ values ($q < 0.02\,\text{Å}^{-1}$) in both the ordered and the disordered materials, attributable to the presence of structures larger than individual micelles, either interconnected networks or jammed star-like micelle packings with strong correlations. These large-scale structures would scatter at inverse length scales corresponding to $q$ values smaller than the range investigated in the experiments reported here, with only the asymptotic power law scattering manifesting as an upturn at low $q$ range investigated in the current study.

**Gel phases in low triblock copolyelectrolyte assemblies.** Intriguing differences in the scattering profiles from di- and triblock copolyelectrolyte self-assemblies appeared at polymer concentrations well below the micelle overlap concentrations (concentrations corresponding to jamming of the unperturbed micelles). At polymer concentrations below 2% by mass, diblock copolyelectrolyte assemblies exhibited weak correlation among the PEC domains, exemplified by a plateauing of the $I(q)$ at low $q$ values ($q < 0.01\,\text{Å}^{-1}$) and indistinct correlation peaks. Contrarily, triblock copolyelectrolyte assemblies exhibited a strong upturn in $I(q)$ at low $q$ values, indicating the presence of large

structures in solution. The scattering intensities also exhibited a shoulder, corresponding to strong correlations between the PEC domains. These correlations decayed with decreasing $\phi$, but their presence at such low $\phi$ was nevertheless surprising.

It was noteworthy that the scattering profiles still coincided over moderate to high $q$ values ($q > 0.05\,\text{Å}^{-1}$), indicating that the morphologies of the individual PEC domains were similar for the di- and triblock copolyelectrolyte assemblies.

Halving the size of the neutral block in both the di- and triblock copolyelectrolytes made the disparities in the scattering patterns from the two macromolecular architectures extremely apparent, as evident in Fig. 1b. Diblock copolyelectrolyte assemblies exhibited scattering patterns characteristic of non-interacting isolated micelles: plateau at low $q$ values ($q < 0.01\,\text{Å}^{-1}$) transitioning to $I(q)$ decaying with a $-4$ scaling exponent at intermediate $q$ values ($0.03\,\text{Å}^{-1} < q < 0.07\,\text{Å}^{-1}$), indicative of scattering from spherical PEC cores, and further $I(q)$ decaying with a $-2$ scaling exponent at high $q$ values ($q > 0.1\,\text{Å}^{-1}$), suggestive of the scattering from the individual polymer chains. Scattering from triblock copolyelectrolyte assemblies, however, exhibited an upturn in the low $q$ region, along with prominent primary and secondary peaks around $q$ values of $0.03\,\text{Å}^{-1}$ and $0.05\,\text{Å}^{-1}$, respectively. These observations, coupled with the $\phi$-invariant peak positions, strongly suggested that triblock copolyelectrolytes self-assembled into gel phases characterized by interconnected networks of PEC domains bridged by the neutral midblocks.

The progression of structures of self-assemblies with $\phi$, as revealed by the scattering results, are depicted by a schematic in Fig. 2. Diblock copolyelectrolytes assemble into dilute solution of star-like micelles at low $\phi$; the micelles jam and eventually assemble into ordered structures with increasing $\phi$. Triblock copolyelectrolytes, contrarily, assemble into gel phases with interconnected networks of PEC domains, even at the lowest $\phi$ investigated in the scattering experiments, with no discernible evidence of flower-like micelles. The finite, non-percolating gels phase separate from the solution at low $\phi$, as was evident from negligible scattering from regions of the solution without any polymer networks. Increasing $\phi$ leads to an expansion of the gel phases until they percolate through the solution, followed by a disorder–order transition of the PEC domains. This is in sharp contrast to the typical assembly of uncharged, amphiphilic ABA triblock copolymers in a B-selective solvent, wherein flower-like micelles form at low polymer concentrations[54], and bridging among the micelles occurs only at significantly high-polymer concentrations. It should be noted that network formation has been reported in isolated cases of telechelic polymer assemblies, wherein triblock copolymers with short solvophobic sticker ends form phase separated interconnected networks, existing in equilibrium with flower-like micelles[55–59].

**Structure characterization of the interconnected gel phases.** Further identification of the structural features of the interconnected networks in the low concentration gel phase was facilitated by extracting the structure factors and PEC domain characteristics via a description of the $I(q)$ data with models

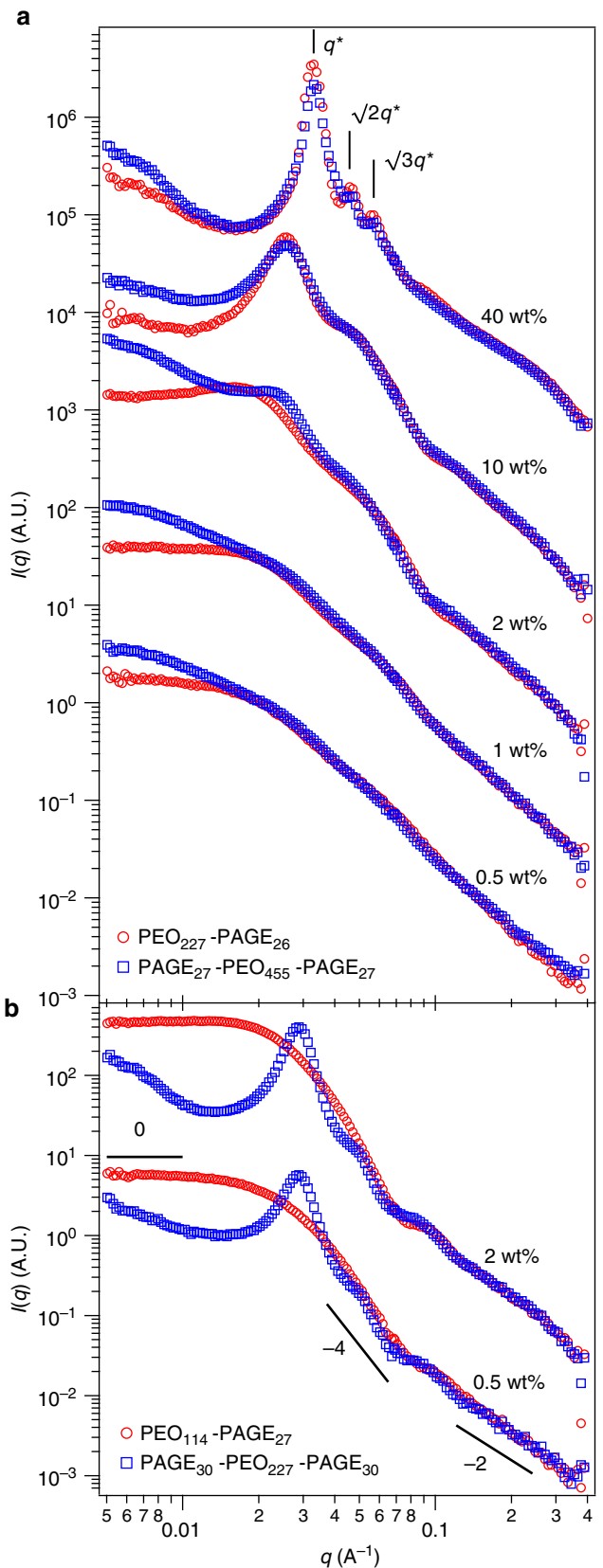

**Figure 1 | Scattering from di- and triblock copolyelectrolyte self-assemblies.** Neutron scattering profiles ($I(q)$ versus wave vector $q$) from self-assemblies comprising oppositely charged diblock (circles) and triblock (squares) copolyelectrolytes at various polymer concentrations. The polyelectrolytes were oppositely charged functionalized forms of (**a**) $PAGE_{27}$-$PEO_{455}$-$PAGE_{27}$ and $PEO_{227}$-$PAGE_{26}$ and (**b**) $PAGE_{30}$-$PEO_{227}$-$PAGE_{30}$ and $PEO_{114}$-$PAGE_{27}$. In (**a**) diffraction peaks corresponding to arrangements of complex domains in a BCC lattice are illustrated for the 40% by mass gel. In (**b**) various power law slopes are indicated by corresponding lines. Error in the data are typically smaller than the symbols and therefore not shown in the figure.

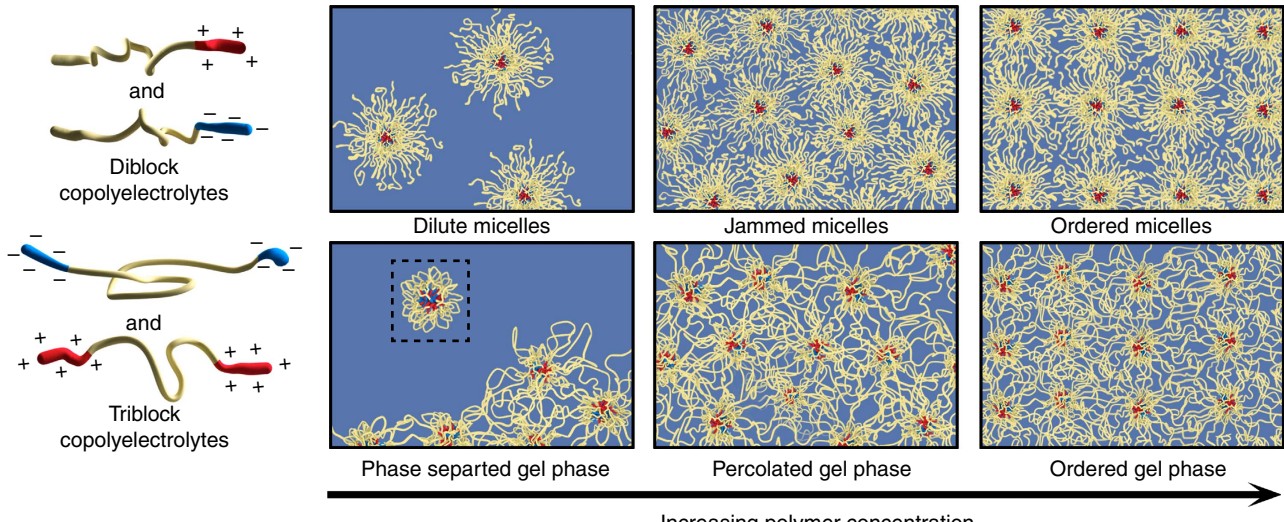

**Figure 2 | Structure evolution of complexation-driven assemblies.** Diblock copolyelectrolytes assemble into star-like micelles. At low polymer concentration, discrete micelles remain suspended in the solution. With increasing polymer concentration, micelles jam, thus forming viscoelastic solids, and eventually order in cubic lattice structures. Triblock copolyelectrolytes are expected to form flower-like micelles at extremely low concentrations (the expected flower-like micelle is shown in the inset), but instead form interconnected gels that phase separate from the solution. Increasing polymer concentrations leads to growth of the networks until they percolate through the solution resulting in viscoelastic solids, followed by a disorder-order transition of the PEC domains. In the corresponding jammed/percolated and ordered phases, PEC domains have very similar size and arrangements, leading to nearly identical scattering patterns.

for block copolymer micelle scattering[46,50]. The details of the model and the fitting procedure can be found in the Supplementary Methods, and the fits are shown in Supplementary Fig. 1. Structure factors ($S(q)$), as obtained from the data fits, are shown in Fig. 3a for both di- and triblock copolyelectrolyte assemblies. While the primary $S(q)$ peaks shifted to right with increasing $\phi$ for the diblock copolyelectrolyte micelles, their $q$-invariance with varying $\phi$ for triblock interconnected gels was evident. The inverse of the primary peaks position ($q^*$) was used to estimate an average inter-domain spacing $d$ ($= 2\pi/q^*$). As shown in Fig. 3b,d, did not vary significantly with varying $\phi$ for the interconnected gels.

An assessment of the neutral block bridge conformations in the interconnected networks was obtained by approximating it to the surface-to-surface distance between the PEC domains. A stretching ratio for the midblock polymers was defined as $SR = (d - 2R_c)/R_{e-e,0}$, with $R_c$ and $R_{e-e,0}$ being the radius of the PEC domains and the equilibrium end-to-end distance of the neutral midblock, respectively. The $SR$ values were found to be approximately around 1.3 irrespective of the polymer size or concentration, as depicted in Fig. 3c, denoting a moderate but consistent ($\sim 30\%$) stretching of the bridging chains. It could be surmised from these observations that the $\phi$-independent structure of the interconnected network gels was strongly influenced by the neutral midblock conformations.

**Molecular dynamics simulations of copolyelectrolyte assemblies.** Insights into the network formation and subsequent phase separation were gained via coarse-grained molecular dynamics simulations. Polymers were simulated as bead-spring chains, with individual beads connected to their neighbours via harmonic springs. One bead represented 10 monomers, for both neutral and charged blocks. The conversion of length scales from simulation to real units was achieved following previously reported atomistic modelling of PEO chains[60]. ABA triblock copolymers were simulated as 28 bead chains,

including two charged end-groups comprising four beads each and representing a $PAGE_{40}$-$PEO_{200}$-$PAGE_{40}$ copolymer. AB diblock copolymers were correspondingly simulated as 14 bead chains (with four bead charged end-groups) and represented the $PAGE_{40}$-$PEO_{100}$ copolymers. The electrostatic and repulsive forces were modelled by the Coulomb interactions making use of the Ewald summation technique and a Lennard–Jones potential, respectively. The strengths of potentials were adjusted to produce aggregation statistics corresponding to experiments. The simulations were carried out at constant volume conditions with an implicit solvent. A Langevin thermostat was employed to regulate temperature in the simulations. Detailed simulation methodology can be found in the Methods section.

Simulations were initiated from random configurations and evolved toward more organized states of PEC self-assembly. Structural analysis of the final equilibrium configurations allowed us to identify the individual PEC cores as well as larger self-assembled structures. Snapshots from the simulation box, shown in Fig. 4a, corroborated with the experimental findings— diblock copolyelectrolytes self-assembled into star-like micelles that were dispersed throughout the simulations box, while triblock copolyelectrolytes assembled into interconnected networks that did not span the simulation box and micro-phase separated into gel aggregates. We did not observe macro-phase separation of the networks, though we believe it was due to relatively small size of the simulation boxes. Larger (though prohibitively computationally expensive) simulation boxes would allow networks to diffuse and merge, thus forming macro-phase separated regions.

Primarily flower-like micelle (oligo-chain aggregate) populations with negligible bridging among them were eventually revealed via simulations at extremely low concentration, inaccessible in laboratory experiments $\phi$ (0.0075% by mass), as shown in Fig. 4b. However, bridging between micelles increased rapidly with increasing polymer concentration, and eventually led to continuous connected gel phase structures at $\phi$, as low

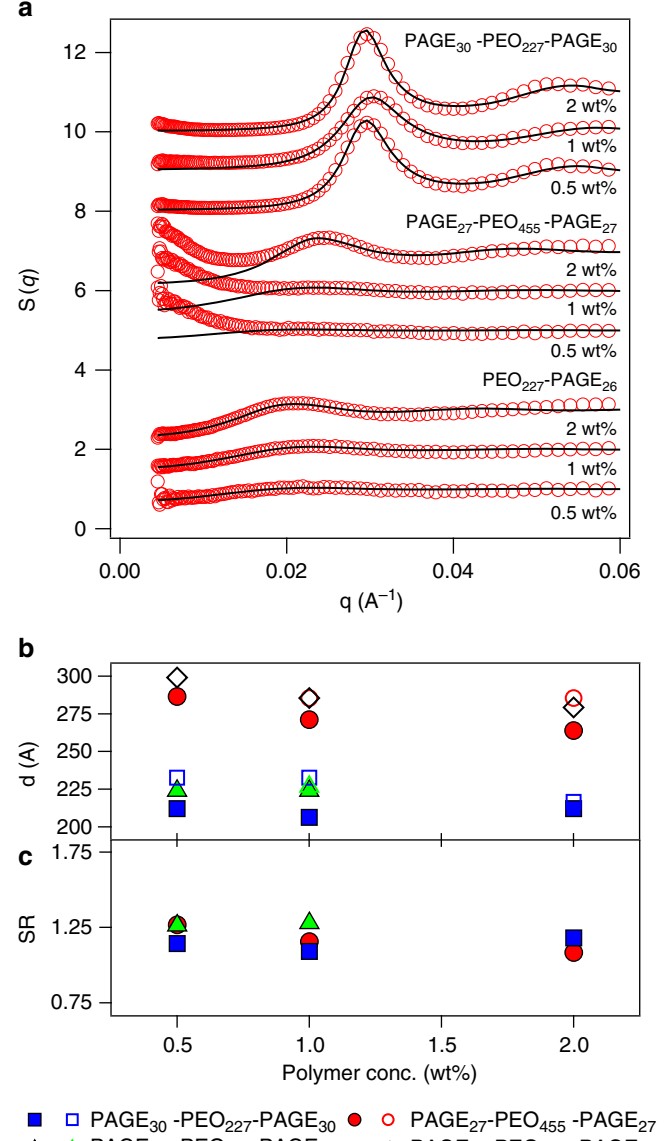

**Figure 3 | Structure of the interconnected gels.** (**a**) Structure factor $S(q)$ for di- and triblock copolyelectrolyte self-assemblies at various polymer concentrations. (**b**) Inter-domain distance $d$, and (**c**) $SR$ ( $= (d - 2R_c)/R_{e-e,0}$) for the neutral midblock as a function of polymer concentration for various triblock copolyelectrolyte gels. In (**a**), successive $S(q)$ plots for different concentrations are shifted vertically by 1 unit each, and collectively by 2 units for different polymer architectures. In (**b**), closed and open symbols denote data obtained from neutron and X-ray scattering, respectively.

as 0.075% by mass. All the flower-like micelles were found assimilated in a single network at $\phi = 0.36\%$ by mass. Interestingly, the PEC domains in the flower-like micelles observed at the lowest concentrations also were composed of fewer chains than those in interconnected networks, as illustrated by the growth of the aggregations numbers with increasing $\phi$, shown in Supplementary Fig. 2, denoting an evolving micellar structure with $\phi$. Bridging and gel phase formation coincided with the stabilization of the aggregation number, denoting the absence of the dilute flower-like micelle solution phase in fully developed triblock copolyelectrolyte assemblies.

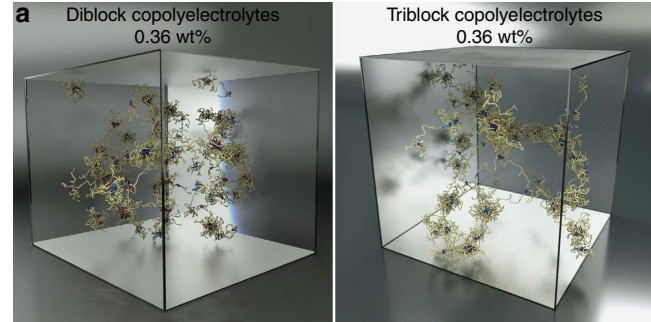

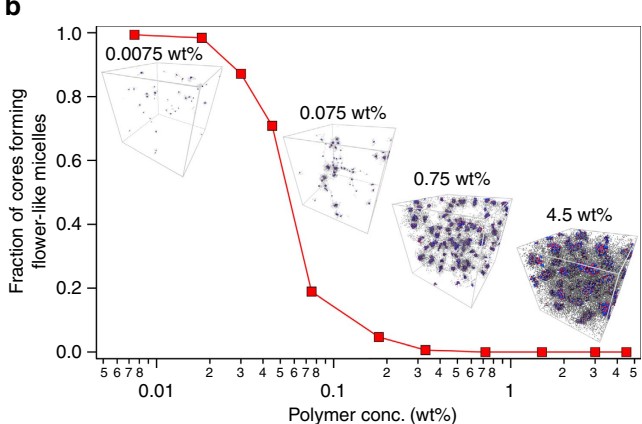

**Figure 4 | MD simulations reveal triblock copolyelectrolyte assembly into interconnected gels.** (**a**) Snapshots of the simulation box showing self-assembled structures comprising oppositely charged di- and triblock copolyelectrolytes. The polycation, polyanion and neutral blocks are depicted by red, blue and yellow coloured beads, respectively. (**b**) Fraction of PEC cores forming isolated flower-like micelles as a function of polymer concentration for triblock copolyelectrolyte assemblies. The fraction approaches a near zero value at $\phi = 0.36\%$ by mass. Insets: Snapshots of the simulation box depicting triblock copolyelectrolyte assemblies at various polymer concentrations. The polycation, polyanion and neutral blocks are depicted by red, blue and grey coloured beads, respectively.

**Neutral block conformation and stretching.** Representative distributions of the end-to-end distance ($R_{e-e}$) of the neutral midblocks for the triblock copolyelectrolyte networks, shown in Fig. 5a, exhibited a clear bimodal distribution with peaks on either side of the $R_{e-e,0}$ corresponding to the looping and bridging midblock chain populations, respectively. At the same time, $R_{e-e}$ distribution for the neutral blocks in diblock copolyelectrolyte micelles showed a unimodal distribution with a maximum at the $R_{e-e,0}$ values (Supplementary Fig. 3). The fraction of midblock chains forming either loops or bridges is depicted in Fig. 5b. At the lowest $\phi$ ( $= 0.0075\%$ by mass), a population of exclusively flower-like micelles corresponded to ~100% midblocks forming loops. However, with increasing $\phi$ the loop fraction decreased (and the bridge fraction increased) rapidly, and a substantial fraction (~30%) of chains formed bridges at $\phi$ as low as 0.2% by mass, leading to formation of interconnected non-percolating gels.

The structure-directing role of the midblock chains in the networks, briefly discussed in Fig. 3c, was exemplified by $\phi$-independent conformations of the midblock chains. The stretching ratio, $SR$, for the bridge-forming midblock chains, defined in the simulations as $R_{e-e}/R_{e-e,0}$ and plotted as a function of $\phi$ in Fig. 5c, indicated nearly $\phi$-independent, moderate

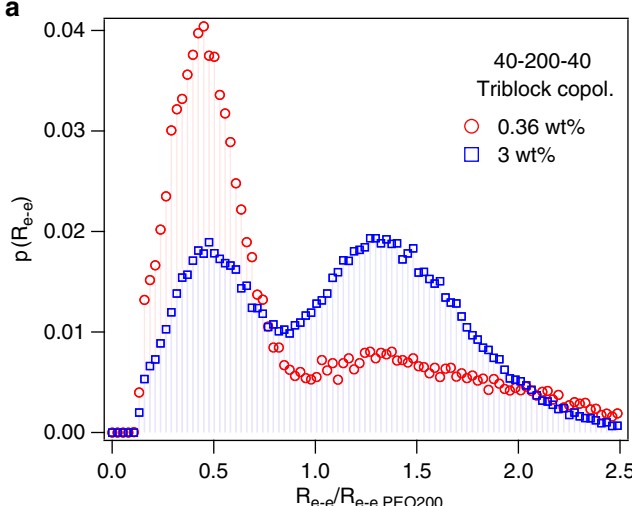

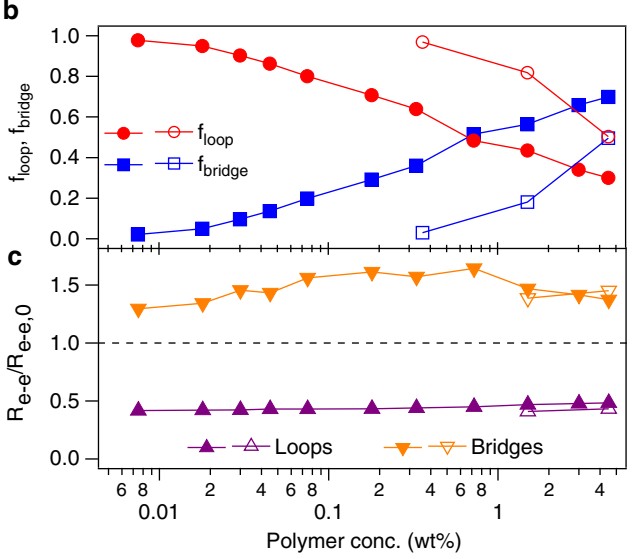

**Figure 5 | Neutral block conformations from MD simulations.**
(**a**) Distribution of the end-to-end distances of the neutral block in the interconnected triblock copolyelectrolyte gels at $\phi = 0.36$ and 3% by mass. (**b**) Fractions of neutral midblocks forming loops or bridges and (**c**) normalized end-to-end distance of the neutral midblocks forming bridges or loops as a function of polymer concentration in the interconnected gels. In (**b**) and (**c**), open symbols correspond to the corresponding data from assembly of uncharged, amphiphilic triblock copolymers.

stretching of the chains. The *SR* determined from simulations is expected to be slightly larger than that estimated from experiments given the approximation of *SR* as $(d - R_c)/R_0$ from the experimental data, and use of harmonic bond potential in simulations. However, excellent agreement between the *SR*, as well as its near invariance with $\phi$ from both experiments and simulations served as strong evidence for the midblock stretching entropy directing the equilibrium network structure. The $\phi$-independent domain sizes and conformations of the midblock chains also reveal a unique evolution of the gel structure with increasing polymer concentration, wherein the networks grew while preserving their internal structure until they percolated through the whole solution.

It is instructive to compare the extent of bridge formation, as well as stretching of the bridge-forming midblocks for the complexation driven assemblies with the corresponding uncharged, amphiphilic ABA triblock copolymers assemblies (shown in Fig. 5b,c as open symbols). Bridging and network formation comparable to triblock PEC networks was observed at ~10-fold higher polymer concentrations in solvophobicity-driven assemblies, as shown in Fig. 5b. Therefore, similar interconnected gel phase networks were obtained in both the cases, although at very different polymer concentrations. Owing to the larger polymer concentrations and consequently larger networks in the latter case, the bridged gels percolated throughout the whole solution and did not phase separate. Further, at comparable $\phi$, complexation driven assemblies exhibited noticeably higher bridge fractions, which could possibly contribute towards the higher fracture strength and moduli of the hydrogels at higher $\phi$. Interestingly, the bridge-forming midblocks were similarly stretched in both forms of assemblies, denoting the similarity of structure-directing roles of the midblocks in both the cases.

An analysis of the $R_{e-e}$ of the loop forming fraction of the midblock chains provides an insight into the physical underpinnings of this enhanced network formation and the ensuing phase separation. Micellization in uncharged, amphiphilic block copolymers is driven by the mutual attraction between the solvophobic blocks, and the entropy penalty for midblock loop formation in the case of triblock copolymers assembling into flower-like micelles is overcome by the energy gains from expulsion of the solvophobic blocks from the solvent as well as their affinity towards each other[54]. Conversely, micellization in the systems presented here was driven by complexation between oppositely charged, fully water-soluble, blocks on distinct chains. Upon complexation of one of the charged endblocks of a chain, there may not be a necessarily strong driving force to recruit the other endblock into the same complex core, aided by the strongly hydrophilic nature of the uncomplexed polyelectrolyte endblocks. Moreover, strong repulsion between the like-charged endblocks upon being brought in the near vicinity of each other, as evidenced by the small $R_{e-e}$ that accompanies the looped midblock populations (Fig. 5c), would provide further hindrance to loop formation of the midblocks unless both the endblocks are completely complexed. These factors may combine to lead to an enhanced propensity of the neutral midblocks to form bridges rather than loops, and thus drive phase separation of the networks.

## Discussion

In summary, spontaneous gel phase formation and an absence of well-dispersed micelle solution phase was observed in complexation driven assemblies of oppositely charged ABA triblock copolyelectrolytes. At low polymer concentrations ($< 2\%$ by mass), the interconnected gels were found to phase separate from the solution, denoting a novel transition to a gel phase, existing between micellization at the extremely low critical micelle concentrations and the disorder-order transition above the micelle overlap concentrations. Molecular dynamics simulations, on samples sufficiently large to demonstrate assembly of multiple cores and formation of larger intercore structures, showed excellent agreement with experimental observation. The short-range repulsion (between the like-charged endblocks) is presumed to drive the long-range attraction by biasing the placement of the like-charged blocks against complexing in the same PEC cores, leading to network formation. This emphasizes the role of the polyelectrolytes in defining the structure of the micelles and their networks in addition to driving the micellization via complexation. At the same time, the

role of neutral blocks goes beyond forming the coronae. The stretching of the bridge-forming neutral blocks strongly influenced the inter-domain spacings, thus directing the structure of the networks. These findings denote a significant advancement in our understanding of polyelectrolyte complexations based assemblies, which were understood until now to proceed analogous to uncharged, amphiphilic block copolymer assemblies.

In combination with excellent encapsulating propensities of PEC assemblies, triblock copolyelectrolyte gel formation at very low concentrations leads to new avenues for development of efficient flocculants for water treatment applications and theranostic carrier-probes. At the same time, our discovery will also aid in establishing the design criteria and tuning the long-term performance of the PEC hydrogels. Greatly enhanced bridging among the micelle cores can be credited for significantly higher equilibrium moduli as well as their slow evolution in triblock copolyelectrolyte gels as compared to diblock copolyelectrolyte gels[47]. Swelling and erosion characteristics of PEC hydrogels will also be notably influenced by the polymer architecture, with gels comprising diblock copolyelectrolytes swelling infinitely and dissolving in the surrounding medium, while hydrogels comprising triblock copolyelectrolytes swelling to a finite extent only. Thus, mixtures of di- and triblock copolyelectrolytes can be employed to prepare hydrogels with precisely controlled moduli, morphology and spacing of cargo and nutrient loaded PEC domains, swelling, and in-media retention times.

## Methods

**Materials.** Di- and triblock copolymers comprising poly(ethylene oxide) (PEO) and poly(allyl glycidyl ether), denoted herein as P(EO-AGE) and P(AGE-EO-AGE), respectively, with varying degrees of polymerizations were synthesized by anionic polymerization, and were subsequently functionalized via thiol-ene click reactions, following previously reported protocols[42,45]. The PEO initiator, AGE monomers, solvents and all the other reagents were obtained from Sigma Aldrich and were used as received.

**Anionic polymer synthesis.** AGE was purified by overnight stirring with calcium hydride (1 g per 25 ml AGE), followed by three freeze-pump-thaw cycles and distillation using a Schlenk line. Reactions were set-up by dissolving either PEO or PEO methyl ether (∼20 g) for triblock and diblock copolymer, respectively, in tetrahydrofuran (THF, ∼200 ml) under a dry argon atmosphere, and then titrated with a solution of potassium naphthalenide dissolved in THF until the solution had a light green coloration. Distilled AGE was then added in appropriate amounts to initiate the polymerization, and the reaction was stirred for 48 h. The polymerization was terminated with degassed methanol (∼10 ml), and precipitated in hexanes, followed by drying in vacuo.

**Polymer functionalization via click chemistry.** *Sodium sulfonate functionalization.* In a 50 ml flask, 1 g of PEO–PAGE copolymer was dissolved in a minimal amount of dimethylformamide. Sodium 3-mercapto-1-propane-sulfonate (4 eq. per alkene) and azobisisobutyronitrile (0.75 eq per alkene) was then added to the solution, and the mixture was sparged for 30 min with N₂. The reaction was then heated at 75 °C while stirring for 12 h. The disappearance of the peaks corresponding to the allyl groups in the ¹H NMR spectra were used to confirm the completion of the reaction. After completion, the contents were transferred to a dialysis bag with molecular weight cutoff of 3,500 Da and dialysed against 4 l MilliQ water for 10 cycles of 8 h each. The polymer product was then obtained by drying the dialysed solution by lyophilization.
*Guanidinium chloride functionalization.* Amine functionalization proceeded analogously to the sulfonate functionalization, except that cysteamine (20 eq. per alkene) was added instead of sodium 3-mercapto-1-propanesulfonate. After 3 cycles of dialysis, the reaction mixture was transferred to a round bottom flask and 1H-pyrazole-1-carboxamidine (4 eq. per alkene) was added to it. The pH was adjusted to 10 using 10 mol l⁻¹ NaOH solution and the reaction mixtures were stirred for 3 days. Finally, the functionalized polymers were obtained by 10 cycles of dialysis and lyophilization.

**Hydrogel preparation.** The functionalized polymers were dissolved separately in MilliQ water to obtain stock solutions with either 5% by mass or 50% by mass polymer. The low concentration assemblies were prepared by mixing the solution

of 5% by mass guanidinium functionalized PEO–PAGE with appropriate amounts of water and then adding the solution of 5% by mass sulfonate functionalized PEO-b-PAGE to achieve a specific polymer concentration $\phi$, defined in w/v units, and equal charge molar ratio. For the high polymer concentration assemblies, 50% by mass stock solutions were used. The concentration and the order of mixing of the polymer stock solutions were found to have minimal impact on the structure of resultant assemblies. The low concentration assemblies were found to be stable for multiple days, and their stability was further probed by subjecting the solutions to overnight sonication. The kinetics of formation of high-concentration structures in these materials systems has been previously shown to be very fast, followed by structural evolution persisting over days owing to evolving packing[47]. The low concentrations structures were therefore expected to form quickly owing to a lack of packing frustration. Visual observations of the phase separated gels in some cases agreed with these expectations, with structures forming over the time-scales of mixing of the polymer solutions.

**Scattering measurements.** Small-angle neutron and X-ray scattering measurements were conducted at EQSANS beamline at Spallation Neutron Source, Oak Ridge National Laboratory and beamline 12-ID-B at Advanced Photon Source, Argonne National Laboratory, respectively. For neutron scattering measurements, samples were sandwiched between quartz plates, sealed and exposed to neutrons for 40 min. Two sample detector distances of 2 m and 4 m were employed, and the data sets collected were merged. Samples were sandwiched between Kapton tapes for X-ray scattering measurements, and the exposure times were limited to 0.1 s. All measurements were conducted at room temperature.

Certain commercial equipment and materials are identified in this paper in order to specify adequately the experimental procedure. In no case does such identification imply recommendations by the National Institute of Standards and Technology (NIST) nor does it imply that the material or equipment identified is necessarily the best available for this purpose.

**MD simulations.** Coarse-grained molecular dynamics simulations, with polymers being represented by bead-spring chains were carried out. The individual beads in each chain were connected to their neighbours via harmonic spring and represented 10 monomers, for both neutral and charged blocks. The electrostatic and repulsive forces were modelled by the Coulomb interactions making use of the Ewald summation technique, and by a Lennard-Jones potential, respectively. Parameters for Lennard–Jones were adapted from conventional explicit water simulation approach ($\rho\sigma^3 = 0.8$). However, water beads were removed and Langevin thermostat with constant volume was applied. Simulation time-step set to 0.025 Lennard–Jones time units, and the systems were equilibrated for $\sim 10^6$ time-steps. The conversion to real concentration was calculated from atomistic simulations[60] and yielded in 180 polymer beads for every $PEO_{200}$ $R_{e-e}^3$ to 1% by mass. The parameters for Coulomb potential was tuned to give aggregation number of PEC core on the order of 20–25 end blocks per core, comparable to experimental observation. Supplementary Fig. 4 shows the relative strength of employed potentials.

**Code availability.** The code generated during and/or analysed during the current study are available from the corresponding author on reasonable request.

**Data availability.** The datasets generated during and/or analysed during the current study are available from the corresponding author on reasonable request.

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

## Acknowledgements

We thank Prof. N.P. Balsara and J. Ting for insightful discussions and H. Acar, A. Marciel, X. Wei, C. Castle and Q. Xu for useful help in the experiments. We acknowledge the gracious support from C. Gao and T.-H. Kang at ORNL, and X. Zuo, B. Lee and J.E. Ernst at ANL during scattering experiments. This work was performed under the following financial assistance award 70NANB14H012 from U.S. Department of Commerce, National Institute of Standards and Technology as part of the Center for Hierarchical Materials Design (CHiMaD). A portion of this research used resources at the Spallation Neutron Source, a DOE Office of Science User Facility operated by the Oak Ridge National Laboratory. This research also used resources of the Advanced Photon Source, a U.S. Department of Energy (DOE) Office of Science User Facility operated for

the DOE Office of Science by Argonne National Laboratory under contract no. DE-AC02-06CH11357.

## Author contributions

S.S. designed and performed the scattering experiments, with help from W.T.H. Polymer synthesis was carried out by S.S., A.L. and D.J.G.; J.M. assisted in the synthesis. M.A. designed and performed computer simulations. All authors contributed to the interpretation of the data. S.S., M.A., A.L., J.J.d.P. and M.T. wrote the manuscript, with inputs from V.M.P. J.J.d.P. supervised the computational work. M.T. supervised the experimental work.

## Additional information

**Competing financial interests:** The authors declare no competing financial interests.

