## [Peer Review File · Nature Communications]

Reviewers' Comments:

Reviewer #1 (Remarks to the Author)

The authors disclose the formation of phase-separated gels in water, using a bicomponent building block approach. Oppositely charged ABA triblock copolymers self-assemble into micelles upon charge neutralisation of the outer A blocks derived from poly(allyl glycidyl ether) functionalised with sulfonate or guanidinium side groups. The middle B block is based on poly(ethylene glycol) and remains solvated by the continuous aqueous phase. In contrast to conventional self-assembly of diblock copolymers with the same block length and composition, discrete micelles could not be observed experimentally with scattering experiments. Using a combined experimental (SAXS and SANS) and theoretical approach (MD simulations) the coauthors provide strong evidence for copolyelectrolyte self-assembly into networks, even at extremely low polymer concentrations. Concentration dependent scattering profiles highlight striking differences for the diblock copolymer spherical assemblies, at weight fractions that the triblock assemblies clearly give rise to higher order morphologies. MD simulations suggest that a dilution to 0.0075wt% of copolyelectrolyte in water is necessary to avoid bridging of triblock copolymers between micellar aggregates and allows the exclusive formation of so-called flower like micelles, where all the B midblocks form loops. The manuscript is well-written, concise, reads like a novel and is very focussed on the scattering data and MD simulations. It will make a very strong impact in the community of functional and dynamic materials that operate in aqueous environments, especially given the huge modularity of the disclosed assembly strategy. I recommend publication with minor changes and suggest the authors expand the discussion slightly based on the following minor comments:

- I suggest the authors include the critical ionic strength at which the triblock copolyelectrolyte complexes start to disassemble. This would help the reader in judging the scope with respect to potential applications. I wonder if the electrostatics driven systems can be reinforced by hydrogen bonding using the phosphonate-guanidinium pairs.
- The energy barriers in charged self-assemblies tend to be large. How important is the sample preparation for the network formation and can the authors exclude kinetic traps? Could the authors give an indication about the time-scale that was needed to reach equilibrium in the scattering experiments.
- The sample preparation reads as follows: "The low concentration assemblies were prepared by mixing the solution of 5 wt% guanidinium functionalized PEO-PAGE with appropriate amounts of water and then adding the solution of 5 wt% sulfonate functionalized PEO-b-PAGE to achieve a specific polymer concentration" Why did the authors decide to add a more concentrated solution of the polyanion to a diluted solution of the polycation. Is the same complex formed if the mixing procedure is reversed, the addition of polycation to the polyanion?

Reviewer #2 (Remarks to the Author)

A well written paper- publish as submitted

Reviewer #3 (Remarks to the Author)

This paper reports a surprising result that complementary ABA and CBC triblock copolymers (where A and C are oppositely charged) form gel-like phase even at extremely low polymer concentrations, where the gel-phase is made of micelle-like blobs. These gel-like phases seem to be percolating, but at the same time maintaining some body-centered-cubic lattice symmetry. This result is in contradiction with the expected flower-like micelles at very low concentrations. As mentioned in the manuscript, such structures are likely to have very interesting bioengineering applications.

As the authors show, with a combination of scattering data and simulations, the bridges of PEO appear to dominate the gelation procedure.

Overall, the paper is well written and it reports a novel result. Nevertheless, there is one major concern. The description by the authors are quite sketchy regarding this concern. This deals with the upturn in intensity at very low scattering angles. The question is whether BCC lattice-like structures and percolating clusters coexist. The authors need to clarify this point, as they mention at several places that there is a phase separation.

Also, the manuscript mentions about the micelle overlap concentration. If I were to take this statement seriously, this overlap concentration is the maximum packing concentration. A clarification is needed.

In Figure 1b, the peaks at 5 and 9 inverse Angstroms need to be discussed (BCC?)

On a minor point, the nature of opposite charges is not obvious from the nomenclature of PAGE. A brief clarification is needed.

Reviewer #1

The authors disclose the formation of phase-separated gels in water, using a bicomponent building block approach. Oppositely charged ABA triblock copolymers ... I recommend publication with minor changes and suggest the authors expand the discussion slightly based on the following minor comments:

Reviewer: *I suggest the authors include the critical ionic strength at which the triblock copolyelectrolyte complexes start to disassemble. This would help the reader in judging the scope with respect to potential applications. I wonder if the electrostatics driven systems can be reinforced by hydrogen bonding using the phosphonate-guanidinium pairs.*

Response: Systematic studies of effects of salt on these systems were previously published. We have modified the text and added those citations in the current manuscript.

We thank the reviewer for pointing out the possibility of reinforcement of the electrostatic interactions with hydrogen bonding using the phosphonate-guanidinium pairs. A similar work for bulk complexes was recently published by our group (Perry et al., Nature Comm., 2015), and we will investigate the suggested approach in future work.

Reviewer: *The energy barriers in charged self-assemblies tend to be large. How important is the sample preparation for the network formation and can the authors exclude kinetic traps? Could the authors give an indication about the time-scale that was needed to reach equilibrium in the scattering experiments.*

Response: The sample preparation protocols were not found to have any effects on the structure of the self-assembled networks. We have included further details of the sample preparation in the Methods section.

Reviewer: *The sample preparation reads as follows: "The low concentration assemblies were prepared by mixing the solution of 5 wt% guanidinium functionalized PEO-PAGE with appropriate amounts of water and then adding the solution of 5 wt% sulfonate functionalized PEO-b-PAGE to achieve a specific polymer concentration" Why did the authors decide to add a more concentrated solution of the polyanion to a diluted solution of the polycation. Is the same complex formed if the mixing procedure is reversed, the addition of polycation to the polyanion?*

Response: The concentration of the stock solutions, and the mixing order were found to have no effect on the structures that were obtained after the self-assembly. We have included these details of the sample preparation in the Methods section.

Reviewer #3

This paper reports a surprising result that complementary ABA and CBC triblock copolymers (where A and C are oppositely charged) form gel-like phase even at extremely low polymer concentrations, where the gel-phase is made of micelle-like blobs. These gel-like phases seem to be percolating, but at the same time maintaining some body-centered-cubic lattice symmetry. This result is in contradiction with the expected flower-like micelles at very low concentrations. As mentioned in the manuscript, such structures are likely to have very interesting bioengineering applications.

As the authors show, with a combination of scattering data and simulations, the bridges of PEO appear to dominate the gelation procedure.

Overall, the paper is well written and it reports a novel result.

Reviewer: *Nevertheless, there is one major concern. The description by the authors are quite sketchy regarding this concern. This deals with the upturn in intensity at very low scattering angles. The question is whether BCC lattice-like structures and percolating clusters coexist. The authors need to clarify this point, as they mention at several places that there is a phase separation.*

Response: The upturn in the scattering intensities was observed only in the solutions wherein structures larger than individual micelles, either star-like or flower-like, existed. The BCC structures and the percolating structures are both large-scale structures, which would produce scattering at length scales corresponding to wave vector values smaller than the range investigated in our experiments. Only the asymptotic power law scattering was therefore observed in our experimental wave vector range, leading to the upturn at low wave vectors. We have modified the text to include these details.

Reviewer: *Also, the manuscript mentions about the micelle overlap concentration. If I were to take this statement seriously, this overlap concentration is the maximum packing concentration. A clarification is needed.*

Response: We would like to point out that the micelle overlap concentration corresponded to the concentration at jamming of the unperturbed micelles. We have added this detail in the text.

Reviewer: *In Figure 1b, the peaks at 5 and 9 inverse Angstroms need to be discussed (BCC?)*

Response: We have added to the pertinent discussion in the text.

Reviewer: *On a minor point, the nature of opposite charges is not obvious from the nomenclature of PAGE. A brief clarification is needed.*

Response: We have modified the text to clarify this point.

REVIEWERS' COMMENTS:

Reviewer #3 (Remarks to the Author):

The authors have satisfactorily addressed my earlier remarks in the revision. I recommend publication as in the current version.

-Muthukumar